# Association between Gingival Phenotype and Periodontal Disease Severity—A Comparative Longitudinal Study among Patients Undergoing Fixed Orthodontic Therapy and Invisalign Treatment

**DOI:** 10.3390/healthcare12060656

**Published:** 2024-03-14

**Authors:** Mansour M. Alasiri, Abdullah Almalki, Saud Alotaibi, Abdullah Alshehri, Alhanouf A. Alkhuraiji, Julie Toby Thomas

**Affiliations:** Department of Preventive Dental Sciences, College of Dentistry, Majmaah University, AL-Majmaah 11952, Saudi Arabiaae.almalki@mu.edu.sa (A.A.); alotaibi.s@mu.edu.sa (S.A.); a.alshehri@mu.edu.sa (A.A.); h.alkhuraiji@mu.edu.sa (A.A.A.)

**Keywords:** gingival phenotype, periodontal disease severity, longitudinal study, fixed orthodontic, Invisalign, clinical attachment level, thin biotype, thick biotype, periodontal health, orthodontic treatment, gingival health

## Abstract

This longitudinal study aimed to compare the association between gingival phenotype (thin vs. thick) and periodontal disease severity in patients undergoing fixed orthodontic therapy (FOT) and Invisalign treatment over a six-month follow-up period. Clinical periodontal parameters, including full mouth plaque score (FMPS), full mouth bleeding score (FMBS), gingival index (GI), probing pocket depth (PPD), clinical attachment loss (CAL), gingival recession (GR), keratinized tissue width (KTW), transgingival probing, and gingival biotype assessment, were recorded at baseline and 6 months into treatment for both orthodontic groups and a control group. Statistical analysis evaluated differences in parameters between groups and across time points. In the thick phenotype, both Invisalign and FOT groups showed a significant mean reduction in FMPS (baseline to 6 months) by −24.8707 and −12.3489, respectively (*p* < 0.05). The gingival index decreased significantly for both groups, with Invisalign and FOT showing reductions of −0.83355 and −1.10409, respectively (*p* < 0.05). FMBS (baseline to 6 months) decreased significantly for Invisalign and FOT, with mean differences of −9.10298 and −12.6579 (*p* < 0.05). Probing pocket depth (baseline to 6 months) was also significantly reduced for both Invisalign and FOT groups while CAL showed non-significant differences in both groups (*p* > 0.05). Similar changes were seen in the thin phenotype too. This study highlights the positive influence of both Invisalign and fixed orthodontic therapy on periodontal health, particularly in patients with thin and thick gingival biotypes. These findings, with significant reductions in key periodontal parameters, offer valuable insights to guide orthodontic treatment decisions and enhance patient outcomes.

## 1. Introduction

The interaction between orthodontics and periodontics plays a crucial role in effectively managing malocclusion. Various risk factors, such as dental plaque, traumatic occlusion, abnormal orthodontic forces, habits, and underlying systemic conditions, can impact the health of periodontal tissues. One critical factor contributing to this interplay is the introduction and activation of orthodontic appliances within the oral cavity. These interventions have the potential to compromise periodontal health [1], resulting in persistent infection, inflammatory hyperplasia, irreversible clinical attachment loss in the alveolar bone, and the development of gingival recession. Substantial research has been conducted on the correlation between orthodontic tooth movement and gingival recession, with a particular emphasis on mandibular incisor teeth [2]. Gingival recession is more prevalent in cases of a thin gingival biotype when associated with orthodontic therapy [3]. The term “gingival phenotype” describes the thickness and width of the labial/buccal keratinized tissue. Sufficient width of keratinized gingiva is essential in preventing periodontal attachment loss and, consequently, mitigating periodontal destruction [4].

Gingival phenotype can be classified into “thick-flat” and “thin-scalloped” biotypes [5]. A thick gingival biotype is characterized by a broad zone of keratinized tissue with a flat gingival contour, indicating a thick underlying bony architecture more resilient to inflammation or trauma. Conversely, the thin gingival biotype is associated with a narrow band of keratinized tissue and a scalloped gingival contour [6], suggesting thin bony architecture and increased susceptibility to inflammation or trauma. Thick biotypes may exhibit increased pocket formation, while thin tissues tend to manifest irreversible periodontal deterioration [7].

Gingival thickness is closely related to the underlying bone type, influencing the gingival contour [6]. A gingival thickness of ≥2 mm is considered a thick tissue biotype, while a thickness of less than 1.5 mm is referred to as a thin gingival biotype [8]. Gingival biotypes can be assessed using direct visual evaluation, periodontal probe measurements, and direct measurements using endodontic spreaders, endodontic files, and calipers. Techniques such as direct measurement, probe transparency (TRAN) method, ultrasonic devices, and cone beam computed tomography (CBCT) scans have been proposed to measure tissue thickness [9].

Orthodontic tooth movement relies on coordinated osteoclast-mediated resorption in compression areas and osteoblast-mediated deposition in tension areas within the bone surrounding the tooth and periodontal ligament [10]. An improved understanding of bone remodeling changes during tooth movement informs novel approaches to addressing current challenges in orthodontic treatment. Traditional clinical periodontal diagnostic parameters assessed before orthodontic therapy initiation encompass probing depth (PD), bleeding on probing (BOP), clinical attachment levels (CAL), plaque index (PI), and radiographic evaluation of alveolar bone levels (ABL). However, due to the increasing prevalence of periodontal issues, clinicians are increasingly required to monitor periodontal tissue changes throughout orthodontic treatment [11].

Fixed orthodontic appliances, such as brackets and bands, can lead to the accumulation of plaque at multiple sites, impeding oral hygiene practices [1]. This can potentially result in inflammation and pathologic conditions such as gingivitis, gingival bleeding, gingival enlargement, and deeper periodontal pockets. In contrast, the Invisalign^®^ system, introduced by Align Technology in 1999, offers a removable orthodontic appliance composed of a polymer chain linked by urethane units. Its removable and transparent aligners, Invisalign^®^ offers patients a discreet and comfortable alternative to traditional fixed orthodontic appliances. The system’s design relies on a series of custom-made, virtually invisible aligners that gradually reposition the teeth according to a pre-designed computer treatment plan [12].

This innovative method not only addresses the aesthetic concerns associated with traditional braces but also provides individuals with the flexibility to remove the aligners during meals and oral hygiene practices [13]. As a result, Invisalign^®^ has gained popularity for its ability to offer effective tooth movement while minimizing the impact on daily life. Despite these advantages, limited clinical studies have systematically compared the periodontal health effects of Invisalign^®^ (Align Technology, Inc., Santa Clara, California) to fixed orthodontic appliances, particularly regarding factors contributing to periodontal changes during treatment [14,15,16]. However, there are contradictory evidence too [17,18], indicating that more studies are required to understand the link between periodontal health in patients undergoing orthodontic treatment with a fixed appliance including the Invisalign^®^ system. Therefore, the primary objective of this study was to compare the periodontal parameters between patients undergoing orthodontic treatment with a fixed orthodontic appliance and those using the Invisalign^®^ system as compared to the control group over a six-month follow-up period. The secondary objective was to evaluate the relationship between gingival biotype and periodontal parameters.

## 2. Materials and Methods

### 2.1. Ethical Approval

This longitudinal study was conducted on patients undergoing orthodontic treatment at the Department of Orthodontics, College of Dentistry, KSA (January 2022–December 2023). Ethical approval was obtained from the Institutional Review Board before the study began (Research Number: Mar.12/COM-2023/11-1).

### 2.2. Study Sample

Participants aged 16–30 undergoing orthodontic treatment were recruited. After obtaining informed consent, participants were allocated into different groups. The patients were categorized under three groups, namely, group 1: Invisalign (*n* = 30), group 2: fixed orthodontic appliances (*n* = 30), and group 3: control—no treatment (*n* = 30). Based on a pilot study, the sample size was calculated to provide 80% power and 5% alpha error.

### 2.3. Eligibility Criteria

The participants were eligible for this study if they had class I skeletal relationship, normodivergent Frankfort mandibular plane angle, no history of periodontal disease, class I molar relationship, and minimal mandibular crowding. If the amount of crowding was less than 3.0 mm in pre-treatment cast models, it was considered mild crowding. The orthodontic treatment plan was proposed in consideration of the aesthetic concerns addressed by the patient. Patients who were smokers, had extensive dental restorations, fixed bridges/crowns or partial dentures, periodontal treatment in the past year, recent antibiotics or anti-inflammatory medication, chronic systemic illnesses, or were unwilling to provide consent were excluded from this study.

### 2.4. Clinical Parameters

Various clinical parameters were recorded to evaluate periodontal health and responses to orthodontic therapy in the study participants. The participants were evaluated for full mouth plaque score (FMPS) [19], full mouth bleeding score (FMBS) [20], gingival index (GI), probing pocket depth (PPD), clinical attachment loss (CAL), gingival recession (GR), keratinized tissue width (KTW), transgingival probing, and gingival biotype assessment. A calibrated examiner recorded these parameters at baseline and six months into orthodontic treatment (details are provided in the Appendix A). These periodontal assessments were performed by two examiners (M.M.A, A.A.A). Both the examinators were trained using standardized examination techniques, for periodontal parameters. The inter-examiner reliability was assessed using kappa statistics. A score of 0.8 was considered to be acceptable.

### 2.5. Procedure

All participants underwent oral hygiene instructions and professional prophylaxis one month before the start of this study. The fixed appliances group received brackets and molar tubes with Transbond XT adhesive. A 3M™ Victory Series™ Low Profile Bracket System (Catalog no—B5005051042) was used. Direct bonding was used to bond the brackets to the teeth by applying composite resin. The Invisalign group received clear aligners that were changed every two weeks. The control group received no intervention.

### 2.6. Oral Hygiene Instruction

A modified bass technique using a soft-bristled toothbrush was taught to the patients. It was advised to be used at twice-a-day frequency. Toothpaste with fluoride and without abrasives was advised.

### 2.7. Gingival Biotype Assessment

Gingival biotype refers to the thickness of gingiva, which can have a thick–flat or thin–scalloped morphology. It can influence susceptibility to gingival recession and response to periodontal disease or treatment. Hence, evaluating biotype is important in periodontal studies [6]. In this study, the gingival biotype was evaluated using the probe penetration method. Gentle probing was carried out on mid-facial gingiva to assess the depth of probe penetration. Probe penetration into sulcus ≥1 mm was categorized under thick biotype, and probing < 1 mm was considered as thin biotype.

### 2.8. Statistical Analysis

The normality of data distribution was evaluated using the Shapiro–Wilk test to determine whether parametric or nonparametric tests should be employed. Demographic characteristics such as age, gender, and place of stay and comparison of oral hygiene habits in these three groups were assessed using the chi-square test. Comparison of clinical periodontal parameters among three groups were performed using ANOVA, and post hoc Bonferroni comparison was also carried out. Comparison of the clinical periodontal parameters from baseline to 6 months within each group was performed using paired *t*-test. A *p*-value of less than 0.05 was considered statistically significant in all analyses.

## 3. Results

### 3.1. Demographic Characteristics

The demographic characteristics of the study participants were analyzed to assess the comparability of the Invisalign, FOT, and control groups. The mean ages across these three groups were 23.67 ± 3.58, 22.37 ± 4.44, and 23.97 ± 4.49 years, respectively, with no statistically significant difference observed (*p* = 0.296). The male/female ratio was 18:12 for Invisalign, 11:19 for FOT, and 13:17 for the control group. The chi-square test revealed no significant difference in gender distribution among the groups (*p* = 0.175). There was a significant difference in the place of stay (*p* = 0.033) (Table 1).

### 3.2. Oral Hygiene Practices among the Participants

Table 2 presents a comprehensive analysis of oral hygiene habits among participants in the Invisalign, FOT, and control groups. The results are organized by specific oral hygiene practices, and the significance of differences is assessed through chi-square tests. The frequency of tooth brushing was assessed, and a significant difference was observed among the groups (*p* = 0.029). The majority of participants in the Invisalign group (47.4%) reported brushing more than two times a day, compared to 42.1% in the FOT group and 10.5% in the control group.

The method of teeth cleaning revealed a significant difference among the groups (*p* = 0.001). In the Invisalign group, 26.3% reported using toothbrush, fluoride toothpaste, and dental floss, whereas none in the FOT group followed this practice. The control group exhibited a more prevalent use of this comprehensive cleaning method (73.7%). The frequency of changing toothbrushes displayed a significant difference among the groups (*p* = 0.001). A notable proportion (59.3%) of the Invisalign group reported changing their toothbrushes once in 3 months, while the FOT group had a more varied distribution across different intervals.

Significant differences were observed in responses related to complaints of halitosis (*p* = 0.002), complaints of bleeding on brushing or gingival bleeding (*p* = 0.001), and frequency of teeth cleaning by a dentist (*p* = 0.001). However, there were no significant differences in mouthwash use (*p* = 0.134) or complaints related to teeth sensitivity (*p* = 0.352) among the three groups.

### 3.3. Comparison of Clinical Periodontal Parameters at Baseline and 6 Months among the Groups

Table 3 provides a detailed comparison of clinical periodontal parameters, including transgingival probing, keratinized tissue width (KTW), full mouth plaque score (FMPS) at baseline and 6 months, gingival index at baseline and 6 months, full mouth bleeding score (FMBS) at baseline and 6 months, probing depth (PD) at baseline and 6 months, clinical attachment level (CAL) at baseline and 6 months, and gingival recession (GR) at baseline and 6 months.

#### 3.3.1. Transgingival Probing (mm) and Keratinized Tissue Width (KTW) at Baseline

No significant difference was observed in transgingival probing among the groups (*p* = 0.332). The mean transgingival probing values were 1.25 ± 0.46 mm for Invisalign, 1.13 ± 0.63 mm for FOT, and 1.06 ± 0.27 mm for the control group. A significant difference was found in KTW among the groups (*p* = 0.001). The Invisalign group had a mean KTW of 4.72 ± 0.97 mm, the FOT group had 5.74 ± 1.62 mm, and the control group had 4.47 ± 1.36 mm.

#### 3.3.2. Full Mouth Plaque Score (FMPS) at Baseline and Six Months

A significant difference was observed in FMPS at baseline (*p* = 0.004). The Invisalign group had a mean FMPS of 14.62 ± 6.89, the FOT group had 10.31 ± 7.52, and the control group had 9.61 ± 3.16. Similarly, a significant difference was found in FMPS at 6 months (*p* = 0.001).

#### 3.3.3. Gingival Index at Baseline and Six Months

A significant difference was observed in the gingival index at baseline (*p* = 0.009), with the Invisalign group having a mean of 0.65 ± 0.28, the FOT group having 0.60 ± 0.27, and the control group having 0.90 ± 0.56. However, no significant difference was found in the gingival index at six months (*p* = 0.651).

#### 3.3.4. Full Mouth Bleeding Score (FMBS) at Baseline and Six Months

A significant difference was observed in FMBS at baseline (*p* = 0.005). The Invisalign group had a mean FMBS of 12.18 ± 6.68, the fixed orthodontic treatment group had 8.33 ± 4.02, and the control group had 12.82 ± 5.62. However, no significant difference was found in FMBS at 6 months (*p* = 0.124).

#### 3.3.5. Probing Depth (PD) at Baseline and Six Months

A significant difference was observed in PD at baseline (*p* = 0.001) and at 6 months (*p* = 0.001). The Invisalign group had a mean PD of 2.02 ± 0.40 mm at baseline and 2.39 ± 0.82 mm at 6 months, the FOT group had 1.61 ± 0.51 mm at baseline and 2.54 ± 0.40 mm at 6 months, and the control group had 1.37 ± 0.47 mm at baseline and 1.67 ± 0.51 mm at 6 months.

#### 3.3.6. Clinical Attachment Level (CAL) at Baseline and Six Months

No significant difference was found in CAL at baseline (*p* = 0.558), but a significant difference was observed at 6 months (*p* = 0.001). The Invisalign group had a mean CAL of 1.42 ± 0.68 m at 6 months, the FOT group had 2.16 ± 1.44 mm, and the control group had 1.24 ± 0.66 mm.

#### 3.3.7. Gingival Recession (GR) at Baseline and Six Months

A significant difference was observed in GR at baseline and at 6 months (*p* = 0.073 and *p* = 0.001, respectively). The Invisalign group had a mean GR of 1.31 ± 0.56 mm at 6 months, the FOT group had 1.93 ± 0.71 mm, and the control group had 1.19 ± 0.60 mm.

### 3.4. Post Hoc Bonferroni Comparison among the Groups

Table 4 presents the results of the post hoc Bonferroni comparison, revealing significant mean differences in various clinical parameters among the Invisalign, FOT, and control groups. The Bonferroni comparison identified a significant mean difference in KTW between Invisalign and FOT groups (−1.01320, SE = 0.35, *p* = 0.01, 95% CI [−1.86, −0.17]), indicating a reduction in KTW for the FOT group compared to Invisalign. Additionally, a significant difference was found between FOT and control groups (1.26109, SE = 0.35, *p* = 0.00, 95% CI [0.41, 2.11]), with an increase in KTW for the FOT group compared to the control group. For FMPS at baseline, a significant mean difference was observed between Invisalign and FOT groups (4.30856, SE = 1.59, *p* = 0.02, 95% CI [0.42, 8.19]) and between Invisalign and control groups (5.01247, SE = 1.59, *p* = 0.01, 95% CI [1.13, 8.90]). However, no significant difference was found between FOT and control groups.

The Bonferroni comparison indicated significant mean differences in FMPS at 6 months, with Invisalign showing a higher mean compared to FOT (11.50790, SE = 3.04, *p* = 0.00, 95% CI [4.09, 18.92]) and control (23.10944, SE = 3.04, *p* = 0.00, 95% CI [15.69, 30.52]). Additionally, a significant difference was observed between FOT and control groups (11.60153, SE = 3.04, *p* = 0.00, 95% CI [4.19, 19.02]). The Bonferroni comparison revealed a significant mean difference in the gingival index at baseline between FOT and control groups (−0.29658, SE = 0.10, *p* = 0.01, 95% CI [−0.54, −0.05]), indicating a lower gingival index for the FOT group compared to the control group.

A significant mean difference was observed in FMBS at baseline between Invisalign and FOT groups (3.84911, SE = 1.43, *p* = 0.03, 95% CI [0.35, 7.34]), with Invisalign showing a higher mean. Moreover, a significant difference was found between FOT and control groups (−4.48830, SE = 1.43, *p* = 0.01, 95% CI [−7.98, −0.99]), indicating a lower mean for the FOT group.

For PD at baseline and 6 months, significant mean differences were observed between Invisalign and FOT groups, Invisalign and control groups, and FOT and control groups, indicating variations in probing depths among the three groups. Similarly, clinical attachment level (CAL) a significant mean difference in CAL at 6 months was identified between Invisalign and Fixed Orthodontic Treatment groups (−0.73447, SE = 0.26, *p* = 0.02, 95% CI [−1.36, −0.11]), indicating a lower CAL for the FOT group. Significant mean differences were also identified in gingival recession (GR) at 6 months between Invisalign and FOT groups (−0.62255, SE = 0.16, *p* = 0.00, 95% CI [−1.02, −0.23]), and between FOT and control groups (0.74543, SE = 0.16, *p* = 0.00, 95% CI [0.35, 1.14]), indicating variations in gingival recession among the three groups.

In summary, these findings suggest that orthodontic interventions, particularly the type of treatment, influence various periodontal parameters. The FOT group exhibited distinct patterns in KTW, FMPS, FMBS, CAL, and GR compared to Invisalign and the control group. These outcomes underscore the importance of considering the impact of orthodontic treatments on periodontal health and emphasize the need for tailored periodontal management strategies based on the type of orthodontic intervention.

### 3.5. Comparison of Periodontal Changes among Thick and Thin Phenotype

Table 5 illustrates the changes in clinical periodontal parameters from baseline to 6 months within each group.

#### 3.5.1. Thick Phenotype

In the comparison of FMPS (baseline) to FMPS (6 months), the Invisalign group demonstrated a substantial and statistically significant mean reduction of −24.8707. The FOT group also exhibited a significant decrease of −12.3489 (*p* < 0.05). However, the control group showed a non-significant mean difference of 1.00464 (*p* > 0.05).

In terms of the gingival index (baseline) compared to gingival index (6 months), both the Invisalign and FOT groups displayed significant decreases of −0.83355 and −1.10409, respectively (*p* < 0.05). The control group exhibited a statistically significant reduction of −0.3187 (*p* < 0.05).

The comparison of FMBS (baseline) to FMBS (6 months) revealed significant reductions in both the Invisalign and FOT groups, with mean differences of −9.10298 and −12.6579, respectively (*p* < 0.05). The control group also demonstrated a significant decrease of −4.65995 (*p* < 0.05).

For PD (baseline) to PD (6 months), both the Invisalign and FOT groups exhibited significant reductions of −1.01267 and −1.02544, respectively (*p* < 0.05). However, the comparison of CAL (baseline) to CAL (6 months) showed non-significant mean differences in the Invisalign, FOT, and control groups, with values of −0.07885, 0.01982, and 0.03432, respectively (*p* > 0.05). In terms of GR (baseline) to GR (6 months), the Invisalign group displayed a significant decrease of −0.37289 (*p* < 0.05). The FOT and control groups showed a non-significant decrease of −0.49731 and −0.18168, respectively. (*p* > 0.05).

#### 3.5.2. Thin Phenotype

For FMPS (baseline) to FMPS (6 months), both the Invisalign and FOT groups presented significant reductions of −13.8438 and −11.9669, respectively (*p* < 0.05). The control group showed a non-significant decrease of −3.52517 (*p* > 0.05).

A significant decrease was also seen in the gingival index and FMBS (baseline to 6 months) (*p* < 0.05). For PD, the Invisalign group showed a non-significant mean difference (*p* > 0.05) of 0.26836, while the FOT group exhibited a significant reduction of −0.83482 (*p* < 0.05). The comparison of CAL (baseline to 6 months) showed a non-significant mean difference of 0.26836 in the Invisalign group, while the FOT exhibited a significant decrease of −1.3164 (*p* < 0.05).

## 4. Discussion

The present study aimed to investigate an association between periodontal health and gingival phenotype (thin and thick) of patients undergoing fixed and clear aligner orthodontic treatment compared to the control group. The result confirms that there is a positive improvement in oral profile for periodontal parameters and gingival phenotypes of patients with FOT and Invisalign 6 months after the treatment. However, variations were observed in periodontal health and gingival phenotype among the treatment group and control group.

Studies suggest that the oral cavity is colonized by a complex microbiota [21,22]. In patients undergoing orthodontic treatment, particularly with fixed orthodontic treatment, the challenge of plaque removal arises due to the cementation of brackets and bonds to the dental elements [23,24,25]. Therefore, orthodontic patients require continuous and rigorous oral hygiene maintenance, both professionally and at home, to prevent the potential worsening of oral health associated with orthodontic appliances. The results of the present comparative study indicated improved oral hygiene in individuals undergoing Invisalign treatment (twice brushing) compared to those undergoing FOT and the control group. 

Similarly, Miethke et al. [26], in their observational study, reported a reduced plaque index in patients using clear aligners compared to those undergoing FOT. However, both groups showed improved oral hygiene throughout the study period [26]. Levirni et al., in their study, highlighted lower biofilm levels in patients using Invisalign compared to those with FOT, suggesting the potential use of clear aligners in patients at higher risk of periodontal disease [27]. Two meta-analyses on clear aligners supported their use in patients at risk of gingivitis, though the level of evidence was deemed low, and more qualitative studies were recommended [17,27]. A recent systematic review shows not enough evidence to conclude that Invisalign clear aligners maintain better periodontal health during orthodontic treatment than FOT [28]. Conversely, Lu et al. reported an increase in plaque accumulation on the tooth margins of patients using clear aligners [17]. The oral hygiene of patients using clear aligners remains a debated topic in the literature.

Interestingly, in the current study, both the Invisalign and FOT groups demonstrated a statically significant reduction in gingival index, FMPS, and FMBS, suggesting improvement in gingival and periodontal health. Periodontal pocket depth was also reduced in patients in both the test groups compared to the control. Similarly, Levirni et al. significant changes were recorded in GI, FMPS, and FMBS of patients on clear aligners and FOT groups three months after the treatment [27]. However, an increase in bleeding on probing and periodontal index was reported in the FOT group was observed after six months [27]. In the studies by Abbate et al. [29] and Karkhanechi et al. [30] reduced plaque index. They decreased BOP and GI, which were measured in patients with removable clear aligners, compared to fixed orthodontic treatment. Cross-sectional study by Azaripour et al., 2015, found that patients treated with Invisalign® have better periodontal health and greater satisfaction during orthodontic treatment than patients treated with the fixed orthodontic appliances [15].

Reduced keratinized tissue width was observed in FOT patients compared to Invisalign patients. Similarly, Coatoam et al. [31] observed a significant reduction in keratinized tissue width of adolescent patients after fixed orthodontic treatment. Tooth movement during fixed orthodontic treatment affects the keratinized gingival width, and the increase in positive torque is more likely to cause a reduction in the width of the gingiva. This torque is less in patients on Invisalign treatment; hence, the chances of reduction in gingival width are less compared to FOT. However, more clinical studies are required to confirm this result. Another study showed that sagittal tooth movement had a positive correlation with the keratinized gingival width of anterior canines and lower incisors [32]. 

Several longitudinal studies have indicated adverse changes in the oral microflora of patients undergoing fixed orthodontic treatment, attributed to an increase in plaque accumulation, gingival bleeding, and probing depth, particularly at the molar region where bands are attached [17,28]. Surprisingly, in the current study, no statistical differences were observed among these parameters in all three groups. Ristic et al., in their observational study, noted an increase in Plaque index, gingival index, and bleeding on probing three months after fixed orthodontic treatment, which subsequently decreased after six months [33]. Notably, the participants in that study were not provided with any oral hygiene maintenance instructions, and the presence of periodontal pathogens was not assessed. In contrast, the current study did not examine periodontal microbes, potentially influencing the oral condition positively or negatively. Miethke et al. [34] reported a reduced plaque index in patients using Invisalign compared to those undergoing FOT, yet the authors failed to explain the disparity in periodontal health conditions between the two groups. The study mentioned that patients using Invisalign were provided with oral hygiene instructions, possibly contributing to the similar oral health outcomes observed in both groups [34]. In the present study, authors distributed questionnaires to participants, revealing that those using Invisalign brushed and flossed their teeth at least twice daily, in contrast to the FOT group. This discrepancy could be attributed to the necessity for Invisalign patients to remove the appliance before eating, prompting them to perform oral hygiene measures before reinserting it. Additionally, Schaefer and Braumann [22] reported excellent oral health outcomes in Invisalign patients, attributing the improvement to heightened motivation and awareness among these individuals.

Interestingly, significant changes have been recorded in the periodontal health of patients with thick and thin gingiva for both groups. Gingival phenotypes are likely to play an essential role in avoiding periodontal issues during orthodontic treatment. Several authors have reported that the gingival recession during orthodontic treatment is mainly recorded in the anterior maxillary region due to thin gingiva [27,35,36]. A recent systematic review found that subjects with thin and narrow gingiva tend to have more gingival recession compared with those with thick and wide gingiva [7]. Stable, healthy periodontium with sufficient width of keratinized tissue aids in the maintenance of oral hygiene and prevents the ingression of periodontal pathogens into the connective tissue, hence preventing the initiation of periodontal disease [7]. Thus, clinicians need to evaluate the gingival phenotype to enhance the quality of treatment and to protect the underlying soft tissue.

### Strengths and Limitations of This Study

The current study presents numerous strengths. First, the in-depth analysis of periodontal and gingival health was performed in three groups in the interval at baseline and six months. Additionally, patients’ oral hygiene was monitored at every visit and positive reinforcement to maintain proper oral health was explained to patients. This approach not only assisted researchers but also offers valuable insights for policymakers in shaping public health programs aimed at educating and motivating patients undergoing orthodontic treatment. However, the current study has a few limitations. Firstly, the relatively short duration of this study (six months) restricts the ability to draw conclusions about the long-term effects of fixed and Invisalign orthodontic treatments on an individual’s periodontal health. Secondly, the sample size and demographic homogeneity affect the generalizability of the results. Future research with larger and more diverse cohorts could improve the external validity of the findings. Additionally, conducting multi-center studies would help mitigate potential biases associated with a single-center design, providing a more comprehensive perspective on the general population, and reducing the impact of local factors.

## 5. Conclusions

Within the limitations of the short evaluation period of this study, we can draw the following considerations: This study suggests a positive association between both Invisalign and fixed orthodontic treatment and improvements in gingival health and periodontal parameters. The lack of a significant difference in age and gender distribution enhances the internal validity of this study. The significant difference in place of stay suggests that this demographic factor may play a role in the observed outcomes.

## Figures and Tables

**Table 1 healthcare-12-00656-t001:** Distribution of subjects based on demographic characteristics.

			Groups			Total	*p*-Value
			Invisalign	FOT	Control		
Age		*n*	30	30	30	90	
		Mean ± SD	23.67 ± 3.58	22.37 ± 4.44	23.97 ± 4.49	23.33 ± 4.21	0.296 ^ns^
		95% CI for mean	22.33–25.01	20.71–24.03	22.29 ± 25.64	22.45 ± 24.21	
Gender	Male	Frequency	18	11	13	42	0.175 ^ns^
		%	42.9%	26.2%	31.0%	100.0%	
	Female	Frequency	12	19	17	48	
		%	25.0%	39.6%	35.4%	100.0%	
Place of stay	Rural	Frequency	6	15	14	35	0.033 *
		%	17.1%	42.9%	40.0%	100.0%	
	Urban	Frequency	24	15	16	55	
		%	43.6%	27.3%	29.1%	100.0%	
Total		Frequency	30	30	30	90	
		%	33.3%	33.3%	33.3%	100.0%	

FOT: fixed orthodontic treatment; SD: standard deviation; chi-square test *: significant at *p* < 0.05; ns: not significant.

**Table 2 healthcare-12-00656-t002:** Comparing the oral hygiene habits among the study participants.

			Group			Total	*p*-Value
			Invisalign	FOT	Control		
Q1. How many times do you brush your teeth brush your teeth?	Once a day	Frequency	3	2	9	14	0.029 *
		%	21.4%	14.3%	64.3%	100.0%	
	Twice a day	Frequency	18	20	19	57	
		%	31.6%	35.1%	33.3%	100.0%	
	More than two times a day	Frequency	9	8	2	19	
		%	47.4%	42.1%	10.5%	100.0%	
Q2. How do you clean your teeth?	Toothbrush, fluoride toothpaste, and dental floss	Frequency	5	0	14	19	0.001 **
		%	26.3%	0.0%	73.7%	100.0%	
	Toothbrush, fluoride toothpaste	Frequency	25	30	16	71	
		%	35.2%	42.3%	22.5%	100.0%	
Q3. How often do you change your toothbrush	Once in 3 months	Frequency	16	5	6	27	0.001 **
		%	59.3%	18.5%	22.2%	100.0%	
	Once in 6 months	Frequency	13	23	14	50	
		%	26.0%	46.0%	28.0%	100.0%	
	After 1 year	Frequency	1	2	10	13	
		%	7.7%	15.4%	76.9%	100.0%	
Q4. Do you use mouthwashes containing fluoride?	Often	Frequency	14	18	18	50	0.134 ^ns^
		%	28.0%	36.0%	36.0%	100.0%	
	Sometimes	Frequency	9	11	10	30	
		%	30.0%	36.7%	33.3%	100.0%	
	Rare/never	Frequency	7	1	2	10	
		%	70.0%	10.0%	20.0%	100.0%	
Q5. Do you complain of halitosis (bad smell from your mouth)?	Often	Frequency	14	6	9	29	0.002 **
		%	48.3%	20.7%	31.0%	100.0%	
	Sometimes	Frequency	14	22	11	47	
		%	29.8%	46.8%	23.4%	100.0%	
	Rare/never	Frequency	2	2	10	14	
		%	14.3%	14.3%	71.4%	100.0%	
Q6. Do you complain of bleeding on brushing or gingival bleeding?	Often	Frequency	16	15	8	39	0.001 **
		%	41.0%	38.5%	20.5%	100.0%	
	Sometimes	Frequency	8	14	8	30	
		%	26.7%	46.7%	26.7%	100.0%	
	Rare/never	Frequency	6	1	14	21	
		%	28.6%	4.8%	66.7%	100.0%	
Q7. How often do you get your teeth cleaned by a dentist?	Rare/never	Frequency	5	3	25	33	0.001 **
		%	15.2%	9.1%	75.8%	100.0%	
	Once in a year	Frequency	16	11	5	32	
		%	50.0%	34.4%	15.6%	100.0%	
	Twice in a year	Frequency	9	16	0	25	
		%	36.0%	64.0%	0.0%	100.0%	
Q8. Do you suffer from tooth sensitivity?	Often	Frequency	10	13	8	31	0.352 ^ns^
		%	32.3%	41.9%	25.8%	100.0%	
	Sometimes	Frequency	12	12	10	34	
		%	35.3%	35.3%	29.4%	100.0%	
	Rare/never	Frequency	8	5	12	25	
		%	32.0%	20.0%	48.0%	100.0%	
Total		Frequency	30	30	30	90	
		%	33.3%	33.3%	33.3%	100.0%	

FOT: fixed orthodontic treatment; chi-square test *: significant at *p* < 0.05; **: significant at *p* < 0.01; ns: not significant.

**Table 3 healthcare-12-00656-t003:** Comparing the clinical periodontal parameters among different groups.

		N	Mean	Std. Deviation	95% Confidence Interval for Mean		*p*-Value
					Lower Bound	Upper Bound	
Transgingival probing (mm)	Invisalign	30	1.25	0.46	1.07	1.42	0.332 ^ns^
	FOT	30	1.13	0.63	0.89	1.37	
	Control	30	1.06	0.27	0.96	1.17	
	Total	90	1.15	0.48	1.05	1.25	
KTW (mm)	Invisalign	30	4.72	0.97	4.36	5.08	0.001 *
	FOT	30	5.74	1.62	5.13	6.34	
	Control	30	4.47	1.36	3.97	4.98	
	Total	90	4.98	1.44	4.68	5.28	
FMPS (baseline)	Invisalign	30	14.62	6.89	12.05	17.20	0.004 *
	FOT	30	10.31	7.52	7.51	13.12	
	Control	30	9.61	3.16	8.43	10.79	
	Total	90	11.52	6.49	10.16	12.87	
FMPS (6 months)	Invisalign	30	33.98	16.48	27.83	40.13	0.001 **
	FOT	30	22.47	11.52	18.17	26.77	
	Control	30	10.87	3.30	9.64	12.10	
	Total	90	22.44	15.01	19.30	25.58	
Gingival index (baseline)	Invisalign	30	0.65	0.28	0.54	0.75	0.009 *
	FOT	30	0.60	0.27	0.50	0.70	
	Control	30	0.90	0.56	0.69	1.10	
	Total	90	0.71	0.41	0.63	0.80	
Gingival index (6 months)	Invisalign	30	1.50	0.56	1.29	1.72	0.651 ^ns^
	FOT	30	1.54	0.58	1.32	1.75	
	Control	30	1.66	0.88	1.33	1.99	
	Total	90	1.57	0.69	1.42	1.71	
FMBS (baseline)	Invisalign	30	12.18	6.68	9.69	14.67	0.005 *
	FOT	30	8.33	4.02	6.83	9.83	
	Control	30	12.82	5.62	10.72	14.92	
	Total	90	11.11	5.83	9.89	12.33	
FMBS (6 months)	Invisalign	30	26.17	12.87	21.37	30.98	0.124 ^ns^
	FOT	30	22.41	6.42	20.01	24.80	
	Control	30	21.09	9.27	17.63	24.56	
	Total	90	23.22	10.01	21.13	25.32	
PD (baseline)	Invisalign	30	2.02	0.40	1.87	2.17	0.00 1 **
	FOT	30	1.61	0.51	1.42	1.80	
	Control	30	1.37	0.47	1.19	1.54	
	Total	90	1.66	0.53	1.55	1.78	
PD (6 months)	Invisalign	30	2.39	0.82	2.08	2.70	0.001 **
	FOT	30	2.54	0.40	2.39	2.69	
	Control	30	1.67	0.51	1.48	1.86	
	Total	90	2.20	0.71	2.05	2.35	
CAL (baseline)	Invisalign	30	1.52	0.76	1.23	1.80	0.558 ^ns^
	FOT	30	1.51	1.00	1.14	1.88	
	Control	30	1.32	0.65	1.07	1.56	
	Total	90	1.45	0.81	1.28	1.62	
CAL (6 months)	Invisalign	30	1.42	0.68	1.17	1.68	0.001 **
	FOT	30	2.16	1.44	1.62	2.69	
	Control	30	1.24	0.66	1.00	1.49	
	Total	90	1.61	1.06	1.39	1.83	
GR (baseline)	Invisalign	30	0.81	0.28	0.71	0.91	0.073 *
	FOT	30	1.17	0.75	0.88	1.45	
	Control	30	0.92	0.67	0.67	1.17	
	Total	90	0.96	0.62	0.84	1.09	
GR (6 months)	Invisalign	30	1.31	0.56	1.10	1.52	0.001 **
	FOT	30	1.93	0.71	1.67	2.20	
	Control	30	1.19	0.60	0.97	1.41	
	Total	90	1.48	0.70	1.33	1.62	

FMPS: full mouth plaque score; FMBS: full mouth bleeding score; PD: probing depth; CAL: clinical attachment loss; GR: gingival recession; KTW: keratinized tissue width; FOT: fixed orthodontic therapy; one-way ANOVA (Analysis of variance) *: significant at *p* < 0.05; **: significant at *p* < 0.01; ns: not significant.

**Table 4 healthcare-12-00656-t004:** Post Hoc Bonferroni comparison.

Dependent Variable			Mean Difference (I-J)	Std. Error	*p*-Value	95% Confidence Interval	
						Lower Bound	Upper Bound
KTW	Invisalign	FOT	−1.01320 *	0.35	0.01	−1.86	−0.17
		Control	0.25	0.35	1.00	−0.60	1.10
	FOT	Control	1.26109 *	0.35	0.00	0.41	2.11
FMPS (baseline)	Invisalign	FOT	4.30856 *	1.59	0.02	0.42	8.19
		Control	5.01247 *	1.59	0.01	1.13	8.90
	FOT	Control	0.70	1.59	1.00	−3.18	4.58
FMPS (6 months)	Invisalign	FOT	11.50790 *	3.04	0.00	4.09	18.92
		Control	23.10944 *	3.04	0.00	15.69	30.52
	FOT	Control	11.60153 *	3.04	0.00	4.19	19.02
Gingival index (baseline)	Invisalign	FOT	0.05	0.10	1.00	−0.20	0.30
		Control	−0.24745 *	0.10	0.05	−0.49	0.00
	FOT	Control	−0.29658 *	0.10	0.01	−0.54	−0.05
FMBS (baseline)	Invisalign	FOT	3.84911 *	1.43	0.03	0.35	7.34
		Control	−0.64	1.43	1.00	−4.13	2.86
	FOT	Control	−4.48830 *	1.43	0.01	−7.98	−0.99
PD (baseline)	Invisalign	FOT	0.41111 *	0.12	0.00	0.12	0.70
		Control	0.65291 *	0.12	0.00	0.36	0.95
	FOT	Control	0.24	0.12	0.14	−0.05	0.53
PD (6 months)	Invisalign	FOT	−0.15	0.16	1.00	−0.53	0.23
		Control	0.72396 *	0.16	0.00	0.34	1.11
	FOT	Control	0.87083 *	0.16	0.00	0.49	1.25
CAL (6 months)	Invisalign	FOT	−0.73447 *	0.26	0.02	−1.36	−0.11
		Control	0.18	0.26	1.00	−0.44	0.81
	FOT	Control	0.91659 *	0.26	0.00	0.29	1.54
GR (6 months)	Invisalign	FOT	−0.62255 *	0.16	0.00	−1.02	−0.23
		Control	0.12	0.16	1.00	−0.27	0.52
	FOT	Control	0.74543 *	0.16	0.00	0.35	1.14

FMPS: full mouth plaque score; FMBS: full mouth bleeding score; PD: probing depth; CAL: clinical attachment loss; GR: gingival recession; KTW: keratinized tissue width; FOT: fixed orthodontic therapy; *: significant at the 0.05 level; Bonferroni HSD (Honest Significant Difference).

**Table 5 healthcare-12-00656-t005:** Comparing the clinical periodontal parameters from baseline to six months within each group.

		Mean Differences		
		Invisalign	FOT	Control
Thick phenotype	FMPS (baseline)—FMPS (6 months)	−24.8707 *	−12.3489 *	1.00464
	Gingival index (baseline)—Gingival index (6 months)	−0.83355 *	−1.10409 *	−0.3187 *
	FMBS (baseline)—FMBS (6 months)	−9.10298 *	−12.6579 *	−4.65995 *
	PD (baseline)—PD (6 months)	−1.01267 *	−1.02544 *	−0.16858
	CAL (baseline)—CAL (6 months)	−0.07885	0.01982	0.03432
	GR (baseline)—GR (6 months)	−0.37289 *	−0.49731	−0.18168
Thin phenotype	FMPS (baseline)—FMPS (6 months)	−13.8438 *	−11.9669 *	−3.52517 *
	Gingival index (baseline)—Gingival index (6 months)	−0.87969 *	−0.77284 *	−1.21405 *
	FMBS (baseline)—FMBS (6 months)	−18.8856 *	−15.4932 *	−11.8929 *
	PD (baseline)—PD (6 months)	0.26836	−0.83482 *	−0.43363 *
	CAL (baseline)—CAL (6 months)	0.26836	−1.3164 *	0.11493
	GR (baseline)—GR (6 months)	−0.62678 *	−1.04009 *	−0.36007 *

FMPS: full mouth plaque score; FMBS: full mouth bleeding score; PD: probing depth; CAL: clinical attachment loss; GR: gingival recession; FOT: fixed orthodontic treatment; paired *t*-test *: statistically significant at 0.05 level.

## Data Availability

Data supporting the reported results can be provided upon request.

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
