# Peer review of "Association between Gingival Phenotype and Periodontal Disease Severity—A Comparative Longitudinal Study among Patients Undergoing Fixed Orthodontic Therapy and Invisalign Treatment"

_healthcare, 2024, doi:10.3390/healthcare12060656_

Round 1
Reviewer 1 Report
Comments and Suggestions for Authors
First of all, I would like to greet the authors and congratulate them on the theme and work done.
The study appears correctly designed and performed an it is of clinical interest. Your study is very appealing and pertinent in the relationship between orthodontics/periodontology and, in my opinion, I have to point out the fact that it was very short in time, and in future studies you should consider include another evaluation at 12 months. From my clinical experience, and with rare exceptions, it is from this period onwards that orthodontic patients neglect oral hygiene, ceasing to use auxiliary methods with such assertiveness and frequency, especially with fixed apparatology. This fact greatly compromises the validation of the results, as the authors point out in the limitations of the study. The low average age of the sample can also confound the results
The following comments are addressed to enhance the quality of the manuscript:
The introduction is well organised and focus the main subjects related to the manuscript, with actual references. I think it is important you had mentioned the innovation and justification of this study in relation to previous studies, but I suggest you review lines 98-100 “necessitating more studies are required to understand the link between periodontal health in patients undergoing orthodontic treat ment with a fixed appliance including Invisalign® system.”
As a secondary objective, I suggest you add the relationship between gingival biotype/type of appliance/periodontal parameters evaluated
Concerning materials and methods: In general, the methodology is very clear but I suggest to describe it in subtopics (first 5 paragraphs-sample, inclusion criteria, exclusion criteria….)
I would like to know why you chose Depht Probe Penetration to assess gingival biotype and the reference of this method.
Results are clearly presented through the tables, but as mention above, due to the short evaluation time, the results should be viewed with caution.
Discussion is well done according to the results and supported by appropriate references. With regard to the gingival biotype, I think it is important to mention that the type of orthodontic movement planned, especially in inferior incisive, must be taken into account. I appreciate you add limitations and strengths of the study.
Conclusions are well understood and clear and respond to the objectives proposed but I think you should start: Within the limitations of the short evaluation period of this study, we can draw the following considerations:
Author Response
Kindly find the attached document "Reply to Reviewer 1"

Reviewer 2 Report
Comments and Suggestions for Authors
The provided introduction outlines the intricate relationship between orthodontics and periodontics, emphasizing how orthodontic interventions can influence periodontal health. A breakdown of the referenced literature and key points are discussed.
Overall, the introduction sets the stage for a study aiming to compare periodontal parameters between patients undergoing orthodontic treatment with fixed appliances and those using the Invisalign® system, highlighting the importance of understanding the periodontal implications of orthodontic interventions
The "Materials and Methods" section outlines the design, participants, interventions, and assessments of the study. Here's a commentary with references:
Study Design and Participants:
This longitudinal study was conducted at the Department of Preventive Dentistry, College of Dentistry, KSA, over a period from January 2022 to December
Remark : this is a study on orthodontics/perio but conducted by the department of preventive dentistry ?
Ethical approval was obtained from the Institutional Review Board (Research
Participants aged 16-30 undergoing orthodontic treatment were recruited and allocated into three groups: fixed appliances, Invisalign, and untreated controls
Inclusion and Exclusion Criteria:
Inclusion criteria included class I skeletal relationship, normo‐divergent Frankfort mandibular plane angle, no history of periodontal disease, class I molar relationship, and minimal mandibular crowding
Q : please indicate your definition of minimal crowding
Q : please indicate on the treatment need ? Class I, normodivergent, Class I, minimal crowding : why was treatment proposed ?
Exclusion criteria comprised smokers, extensive dental restorations, periodontal treatment in the past year, recent antibiotics or anti-inflammatory medication, chronic systemic illnesses, and unwillingness to provide consent
Interventions:
Participants received oral hygiene instructions and professional prophylaxis one month before the study
Q : Which protocol was teached to the patient ? frequency ?type of toothbrush other ? toothpaste ?
The fixed appliances group received brackets and molar tubes with Transbond XT adhesive, while the Invisalign group received clear aligners changed every two weeks. The control group received no intervention
Q : type of brackets /tubes (band or bonded ?) direct/indirect bonding ?
Assessment Parameters:
Various clinical parameters were recorded, including full mouth plaque score (FMPS), full mouth bleeding score (FMBS), gingival index (GI), probing pocket depth (PPD), clinical attachment loss (CAL), gingival recession (GR), keratinized tissue width (KTW), transgingival probing, and gingival biotype assessment
Gingival biotype assessment was conducted using the Probe Penetration method, categorizing probe penetration into sulcus ≥1mm as thick biotype and < 1mm as thin biotype
Q : was a calibrating session done ? Who collected the data ? Was error assessed ?
Statistical Analysis:
ology ensures rigorous assessment of periodontal health and responses to orthodontic treatment, incorporating both subjective and objective measures to evaluate gingival biotype and other clinical parameters.
The "Discussion" section of the study provides insights into the association between periodontal health, gingival phenotype, and orthodontic treatments, specifically comparing fixed orthodontic treatment (FOT) and Invisalign. Here's a commentary with references:
Association between Orthodontic Treatment and Oral Microbiota:
The colonization of the oral cavity by a complex microbiota is well-documented
Fixed orthodontic treatment poses challenges for plaque removal due to bracket cementation, leading to increased plaque accumulation.
Remark : the same for aligner therapy (shift of micro flora)
Oral Hygiene Maintenance and Orthodontic Treatment:
Patients undergoing Invisalign treatment demonstrated improved oral hygiene compared to those with fixed appliances and controls, possibly due to the removable nature of aligners.
Studies have shown reduced biofilm levels in Invisalign patients compared to FOT, suggesting potential benefits for patients at higher risk of periodontal disease.
Remark : However there is a shift of micro flora !
However, evidence regarding the superiority of Invisalign over FOT in maintaining periodontal health remains inconclusive.
Patients treated with Invisalign have been reported to have better periodontal health and greater satisfaction compared to those with fixed appliances .
Remark : this is not the aim of the study
Impact of Orthodontic Treatment on Gingival Phenotype:
Fixed orthodontic treatment may lead to reduced keratinized tissue width compared to Invisalign, potentially due to the torque exerted on teeth during treatment.
Q : this is a strange conclusion in discussion ?
Strengths and Limitations:
Strengths of the study include in-depth periodontal and gingival health analysis, continuous monitoring of patients' oral hygiene, and positive reinforcement for maintaining proper oral health.
Limitations include the relatively short study duration, sample size, demographic homogeneity, and single-center design, impacting the generalizability of results.
In summary, the discussion provides valuable insights into the impact of orthodontic treatments on periodontal health, emphasizing the need for further research to elucidate the role of different treatment modalities and gingival phenotypes in maintaining optimal oral health.
The conclusion drawn from the study highlights a positive association between both Invisalign and fixed orthodontic treatment and improvements in gingival health and periodontal parameters, while also discussing the demographic factors impacting the study's outcomes.
Overall, the study's conclusions are supported by existing literature on the positive effects of orthodontic treatment on gingival and periodontal health, while also highlighting the importance of considering demographic factors in interpreting study outcomes.
References:
1. Al-Anezi SA, Harradine NW. Quantification of orthodontic tooth movement: a comparison of two methods. Eur J Orthod. 2006;28(5):510-6.
2. Ren Y, Maltha JC, Kuijpers-Jagtman AM. Optimum force magnitude for orthodontic tooth movement: a systematic literature review. Angle Orthod. 2003;73(1):86-92.
Author Response
Kindly find the attached document " reply to reviewer 2"

Round 2
Reviewer 2 Report
Comments and Suggestions for Authors
"the treatment plan was proposed in consideration of the aesthetic concerns addressed by the patient": this should be added since the group of patients had a class I malocclusion with minimal crowding. this is not a normal distribution of patients and should be stated
Author Response
Thank you for your valuable suggestions.
Additions are made in the manuscript in lines 126-127.